# A BAC-guided haplotype assembly pipeline increases the resolution of the virus resistance locus *CMD2* in cassava

Luc Cornet[1*†], Syed Shan-e-Ali Zaidi[1†], Jia Li[2], Yvan Ngapout[3,9], Sara Shakir[1,3,9], Loic Meunier[4], Caroline Callot[5], William Marande[5], Marc Hanikenne[6], Stephane Rombauts[2], Yves Van de Peer[2,7,8] and Hervé Vanderschuren[1,3,9*]

[†]Cornet Luc and Syed Shan-e-Ali Zaidi contributed equally to this work.

*Correspondence:
luc.cornet@uliege.be; herve.vanderschuren@kuleuven.be

[1] Plant Genetics and Rhizosphere Processes Laboratory, TERRA Teaching and Research Center, Gembloux Agro-Bio Tech, University of Liège, Gembloux Agro-Bio TechGembloux, Belgium
Full list of author information is available at the end of the article

## Abstract

**Background:** Cassava is an important crop for food security in the tropics where its production is jeopardized by several viral diseases, including the cassava mosaic disease (CMD) which is endemic in Sub-Saharan Africa and the Indian subcontinent. Resistance to CMD is linked to a single dominant locus, namely *CMD2*. The cassava genome contains highly repetitive regions making the accurate assembly of a reference genome challenging.

**Results:** In the present study, we generate BAC libraries of the CMD-susceptible cassava cultivar (cv.) 60444 and the CMD-resistant landrace TME3. We subsequently identify and sequence BACs belonging to the *CMD2* region in both cultivars using high-accuracy long-read PacBio circular consensus sequencing (ccs) reads. We then sequence and assemble the complete genomes of cv. 60444 and TME3 using a combination of ONT ultra-long reads and optical mapping. Anchoring the assemblies on cassava genetic maps reveals discrepancies in our, as well as in previously released, *CMD2* regions of the cv. 60444 and TME3 genomes. A BAC-guided approach to assess cassava genome assemblies significantly improves the synteny between the assembled *CMD2* regions of cv. 60444 and TME3 and the CMD2 genetic maps. We then performed repeat-unmasked gene annotation on CMD2 assemblies and identify 81 stress resistance proteins present in the CMD2 region, among which 31 were previously not reported in publicly available *CMD2* sequences.

**Conclusions:** The BAC-assessed approach improved *CMD2* region accuracy and revealed new sequences linked to virus resistance, advancing our understanding of cassava mosaic disease resistance.

## Background

Cassava (*Manihot esculenta* Crantz) is an important food security crop in the tropics [1]. It has an annual production of about 315 million tonnes and is cultivated on over 30 million hectares. Cassava is a particularly important crop in Africa where 65% of the global

cassava production takes place (FAOSTAT). Cassava roots are rich in carbohydrates and serve as a critical food source for over 800 million people, particularly in rural areas where it also provides income. Consequently, cassava ranks as one of the major staple foods. Cassava's ability to grow in nutrient-poor soils and withstand drought makes it a lifeline for smallholder farmers with limited access to inputs like irrigation, fertilizers, and pesticides [2–4]. Despite its high yield potential [5–8] and its anticipated improved performance under climate change conditions [7, 9, 10], cassava production remains severely constrained by several biotic and abiotic stresses. In Sub-Saharan Africa, two viral diseases, namely cassava mosaic disease (CMD) and cassava brown streak disease (CBSD), are widely distributed and they severely limit cassava production [11]. CMD, caused by viruses in the Geminiviridae family, and CBSD, linked to Potyviridae family viruses, present a dual challenge. These diseases have devastating effects on yield, endangering food security and the livelihoods of farmers. Cassava is propagated through stem cuttings, which facilitates the spread of CMD and CBSD, compounding the problem. Recent studies have indicated that cassava virus diseases are spreading rapidly within and beyond Africa, emphasizing the urgent need for developing virus-resistant cassava varieties [12, 13]. Resistance to CMD is linked to a single dominant locus, the so-called *CMD2* locus, which was initially identified in CMD-resistant cassava landraces collected across West Africa and mapped on chromosome 8 [14–17]. *CMD2*-based resistance has been extensively used to introgress CMD resistance in cassava breeding lines and released varieties [18–20]. Important research programs such as NextGen Cassava are now bringing cassava breeding into a new era [21], taking advantage of high-throughput sequencing (HTS) technology to provide breeders and researchers with fully annotated reference cassava genomes along with single nucleotide polymorphism (SNP) information [14, 22–24]. While HTS has been instrumental to generate many plant genome assemblies in a time and cost-effective manner, complex repeats and haplotype heterozygosity have remained major sources of assembly errors in released genomes [25–27]. The accurate assembly of plant genomes, which can contain up to 85% of repetitive elements, remains particularly challenging [28]. The recent development of long-read sequencing has opened new opportunities to improve the resolution of complex repeat-rich genomic regions [26, 29]. In this context, the sequencing of ultra-long reads (ULR) using the Oxford Nanopore Technology offers new opportunities to improve the assembly of complex genomic regions as previously demonstrated for the resolution of repeat-rich regions of the human genome [30]. Cassava has one of the most repetitive plant genomes as repeats are estimated to account for 61% of the total genome sequence [31]. Here, we report the release of the two haplotype sequences of *CMD2* genomic region, sequenced using ULR, from two cassava genotypes contrasting for CMD resistance. A BAC-based approach was established to independently assess the quality of the assemblies in the repeat-rich *CMD2* region and to select the best assembled *CMD2* region among multiple assemblies. A comparison with the available genetic maps of the *CMD2* region shows that the *CMD2* region assembled by our ULR-based approach had a significantly better synteny and contiguity than *CMD2* regions from previously released cassava genomes. We subsequently performed gene annotation and identified the presence of additional resistance genes within our newly assembled *CMD2* regions.

## Results

In order to optimize the assembly of the cassava genomes, we implemented an assessment by bacterial artificial chromosome (BAC) mapping. We first generated BAC libraries of cassava cv. 60444 and TME3, named Mes-60444 and Mes-TME3, that contained 41,472 clones (108 384-well microplates) and 55,296 clones (144 384-well microplates), respectively. We then screened BAC libraries using high-density filter hybridization with radioactively labeled probes. *CMD2*-specific probes were designed based on simple-sequence repeat (SSR) [32–35] and genome-wide SNP markers [15, 16] (Additional file 1: Table S1). After several rounds of probe design—BAC library screening—BAC-end Sanger sequencing—selection of overlapping BACs—BAC PacBio sequencing, we identified 13 and 16 BACs of cv. 60444 and TME3 covering the *CMD2* region, respectively. These BAC sequences from repeat-rich regions were subsequently used to assess the accuracy of ULR-based genome assemblies and provided a powerful parameter to select the best performing assembler (Fig. 1A).

Ultra-long read (ULR) sequencing [30] of HMW cassava DNA on MinION flow cells generated 39,525 Mb (6,221,230 reads) and 41,340 Mb (6,549,320 reads) sequencing data for cv. 60444 and TME3, respectively, with 22,782 reads above 8 kb of length. Three multiple long read assemblers were used to perform the assemblies and assess the impact of increasing read lengths on the quality of the *CMD2* region assembly: *wtdbg2* [37] with 6 different minimal read lengths from 0.3 to 10 kb, *CANU* [38] with all reads and only reads longer than 8 kb, and *Flye* [39, 40] with reads longer than 8 kb. We implemented a method computing the contiguity of BAC sequences in the assembled *CMD2* regions to determine the quality of the assembled *CMD2*. Each assembly was thus assessed by BAC mapping. The contiguity values ranged from 44.9 to 81.45% of BAC mapping on the assemblies (Additional file 1: Table S2), showing an important variation depending on the read length and the assembler used. Our results indicated that mapping of high-accuracy BACs to the assembled genome could be used as a parameter to assess the quality of the assembled *CMD2* regions. Assemblies generated using Flye [39, 40] and a cut-off threshold for reads smaller than 8 kb displayed the best contiguity with BAC sequences despite a decrease in genome coverage (Additional file 1: Table S2). These assemblies mapped 81.45% and 80.2% of contiguous BAC sequence, for cv. 60444 (GCA_963409065.1 Cornet) and TME3 (GCA_963409055.1 Cornet), respectively (Fig. 2), against 73.15% and 61.08% for the previously released cassava genomes of cv. 60444 (GCA_003957885.1 Kuon) and TME3 (GCA_003957995.1 Kuon), respectively [41]. To assess the quality of the *CMD2* region with an independent approach, we took advantage of the publicly available genetic markers for the *CMD2* region [15, 16]. Markers indicated the presence of two haplotypes of the *CMD2* region in our two genome assemblies (Fig. 1B). The length of the *CMD2* region (Fig. 1B) in these haplotypes varied moderately when compared to the length of *CMD2* (1.88 Mb) estimated on marker mapping reported by Rabbi and colleagues [15]: 2.18 Mb (1.16x) and 2.31 Mb (1.23x) for cv. 60444 (GCA_963409065.1 Cornet) and 2.22 Mb (1.18x) and 2.25 Mb (1.19x) for TME3 (GCA_963409055.1 Cornet). Importantly, our assemblies had the same marker order as in the genetic map (Fig. 1B). When analyzing the markers in the *CMD2* region from the public released cassava genomes [41], we noticed they contain a single genomic region

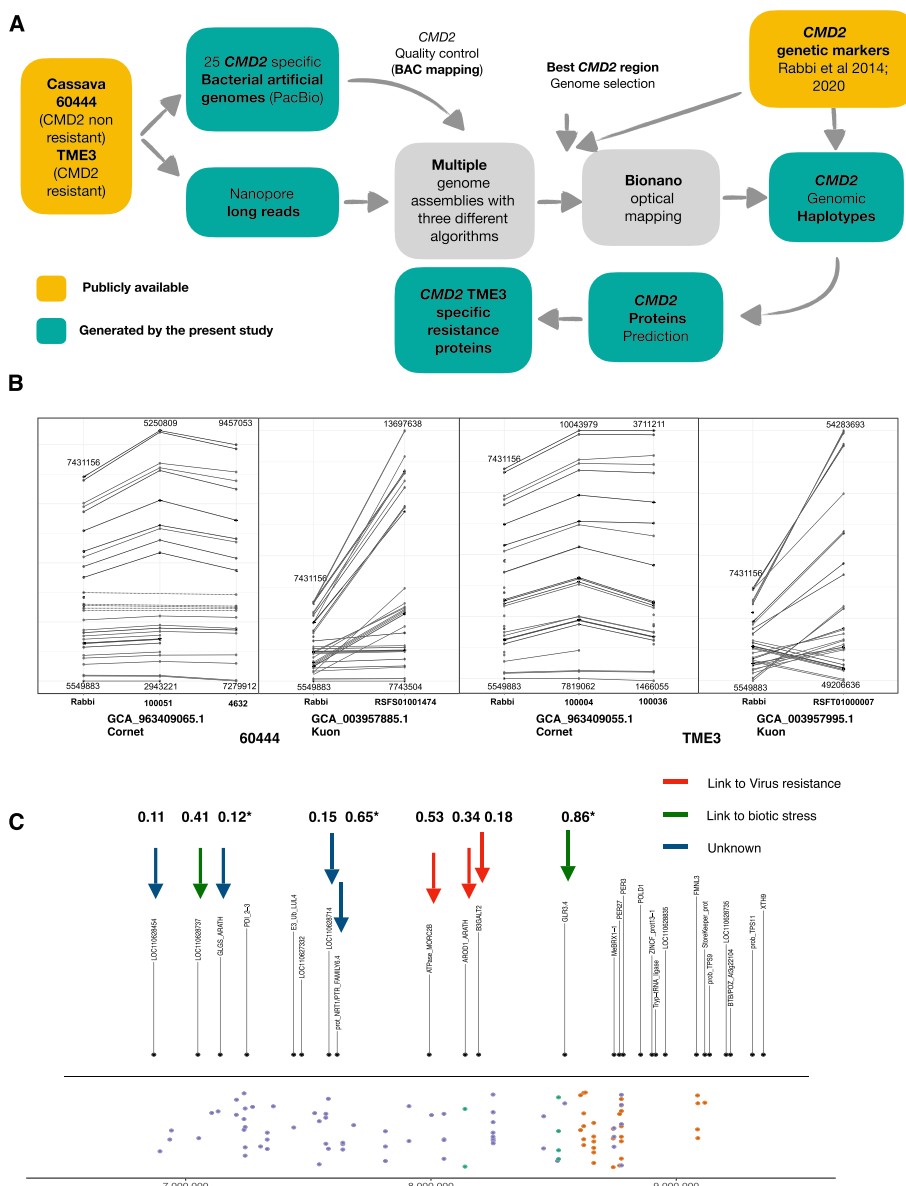

**Fig. 1** **A** Pipeline of the study. **B** Mapping of 64 genetic markers. The markers from Rabbi et al. [15] were mapped by blastn. First hits of each marker were plotted with ggplot2 according to their coordinate. **C** Markers and gene mapping. The extended CMD2 locus in chromosome 12 of the AM560-2 V8 reference genome. The black dots (above the horizontal black line) show the alignment positions of genes of interest that can be found in the *CMD2* from the genome AM560-2 V8 reference genome. The colored dots (below the horizontal black line) indicate various molecular markers associated with the CMD resistance. The green dots indicate classical markers (RFLP and SSR markers) previously published by Akano et al. [32], Lokko et al. [33], Okogbenin et al. [34], and Okogbenin et al. [36]. Orange dots indicate *CMD2* SNP markers published by Rabbi et al. [17], and violet dots indicate markers published by Wolfe et al. [16]. The x-axis of the plot indicates the base pair (bp) position on chromosome 12 of AM560-2. Putative resistance proteins with prPred prediction above 0.11 are indicated by arrows (prPred prediction are indicated above each arrow)

for *CMD2* in which the two haplotypes are merged, with a consistent increase of the theoretical length calculated by genetic mapping [15] [i.e., 5.95 Mb (3.16x) for cv. 60444 (GCA_003957885.1 Kuon) and 5.08 Mb (2.70x) for TME3 (GCA_003957995.1 Kuon)].

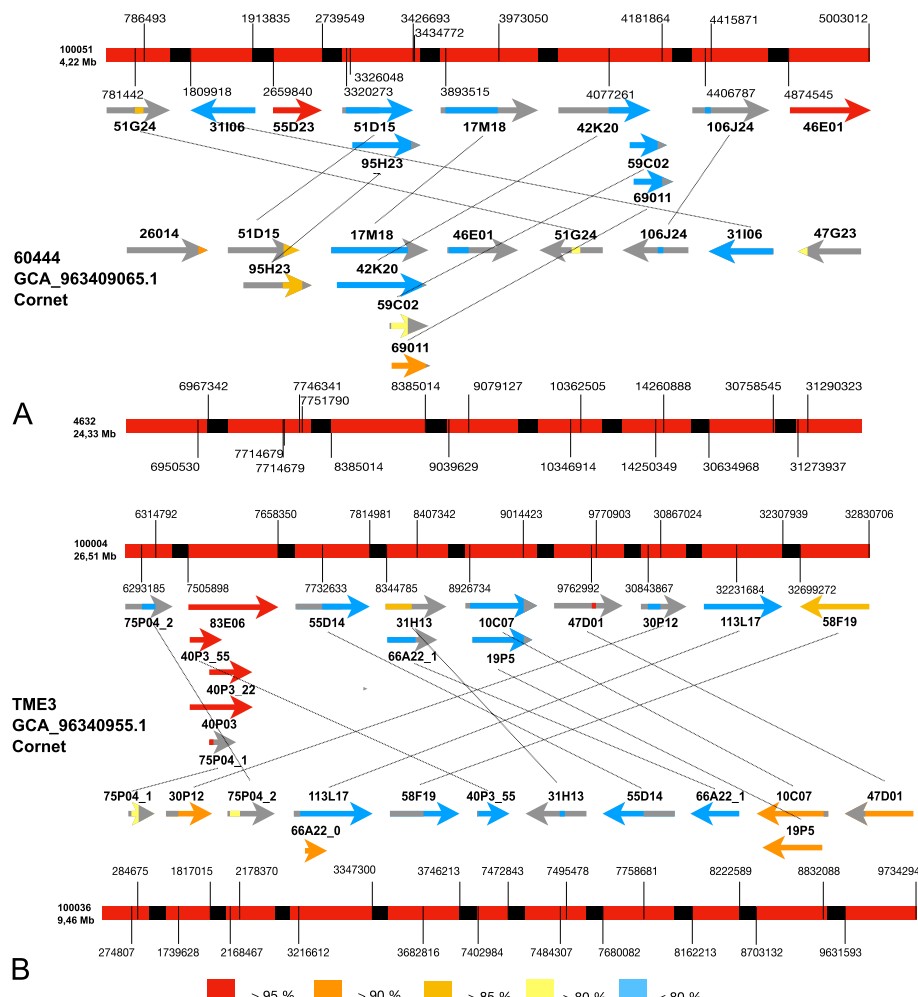

**Fig. 2** BAC mapping on haplotypes. **A** cv. 60444 haplotypes. **B** TM3 haplotypes. Haplotypes are represented in red with break in black. Coordinates of the BAC mapping regions are indicated on the haplotypes. BACs are represented by arrows and non-mapping regions are represented in gray. Identity of the BAC mapping is represented by colors of the BAC

Moreover, the *CMD2* region of the publicly released genomes contained 48 inversion events as compared to the *CMD2* genetic map (Fig. 1B).

Our assembly process, which was optimized for the contiguity of the *CMD2* region, also generated whole-genome assemblies. Although our genome assemblies produced from ULR displayed a more accurate assembly and contiguity for the *CMD2* region, the completeness scores (72.9% for cv. 60444, 95.8% for TME3) as indicated by BUSCO [42] (Additional file 1: Table S3) remained lower than previously released public assemblies.

Annotation of our two newly produced *CMD2* haplotypes from the resistant landrace (TME3) indicated the presence of 79 genes for *CMD2* region of contig 100,004 and 78 genes for *CMD2* region of contig 100,036, respectively (same id used in Fig. 1B). In the cv. 60444, the *CMD2* haplotypes contained 90 genes for contig 4632 and 93 genes for contig 100,051, respectively. Among the 157 genes from the *CMD2* region of TME3, 81 genes had a non-null probability of contributing to CMD2 resistance (see Methods).

The 81 genes were then searched for in the public 60444 (GCA_003957885.1 Kuon) and TME3 (GCA_003957995.1 Kuon) genomes from the NCBI (Additional file 1: Table S4). Thanks to the better resolution of our *CMD2* region assemblies, 31 additional genes (out of 81) were present in our *CMD2* haplotypes but were absent in the publicly available TME3 (GCA_003957995.1 Kuon) genome. Twenty-one of these 31 genes, present on TME3 (GCA_963409055.1 Cornet), did not have reported expression in the cv. 60444 Isoseq dataset [41] (Additional file 1: Table S4) and might represent additional candidate genes for virus resistance potentially complementary to the previously reported *MePOLD1* gene associated with CMD resistance [43]. We analyzed the expression profile of the 81 genes in a public IsoSeq dataset for cv. 60444 and TME3 [41]. Nine genes, 3 of them absent from the TME3 (GCA_003957995.1 Kuon) genome, were expressed in TME3 and non-expressed or absent in 60444 (Fig. 1C). Among these genes, 5 have been reported to be associated with biotic stress resistance, including mosaic virus, in other plants such as *Arabidopsis* [44] or tobacco [45] (Fig. 1C; Additional file 1: Table S5).

## Discussion

The higher BAC sequence mapping values, preserved genetic marker order, and presence of two haplotypes make the newly assembled genomes (GCA_963409065.1 Cornet, GCA_963409055.1 Cornet) the best currently available for landraces cv. 60444 and TME3. A further characterization of the assemblies generated during this study has revealed assemblies with higher completeness had been generated, but they were not selected due to their lower BAC mapping percentage. For instance, the assembly of cv. 60444 with CANU [38] using all reads had a completeness score of 98% but a BAC mapping score of 67.46% (Additional file 1: Table S2). While our results clearly indicate that ONT reads used with BAC-guided assessment of the assemblies provide a significant improvement of genomic regions with high levels of repetitive elements, the use of long reads with high base-level resolution [46] could help achieving both high completeness of the genome and high accuracy in repeat-rich regions.

We identified 81 genes with a non-null probability of contributing to CMD2 resistance making them of high importance to advance the characterization of CMD2 resistance in cassava. Previous breeding work demonstrated that *CMD2*-based resistance is linked to a single dominant locus [15, 16, 32]. *CMD2*-based resistance has recently been shown to co-segregate with single nucleotide polymorphisms in the *MePOLD1* gene [43]. Noticeably an analysis focusing on 101 cassava markers associated with the *CMD2*-based resistance indicated that the *MePOLD1* gene was not located in the *CMD2* region where the majority of the markers mapped (Fig. 1C, Additional file 1: Figs. S1 and S2). Additional wet lab experiments would be required to assess the possibility of multiple biotic stress resistance genes contributing to the CMD2 resistance. Because the IsoSeq transcriptome profiling was not performed in virus-infected plants, we cannot exclude that some of the 81 resistance genes reported here would have a better significance under viral conditions than the one highlighted above. Indeed, nine of those 81 genes appear expressed in virus-free TME3 while they remain silent in virus-free cv. 60444 using IsoSeq data. Therefore, transcriptome profiling of selected contrasting cassava accessions under viral infection as well as gene functional validation using CRISPR technologies [47] will

help providing further insights into the functions of the annotated genes located in the *CMD2* region.

## Conclusions

Our BAC-assessed approach to assemble complex genomes and genomic regions provides a framework to improve the plant genome assemblies that have been released so far. The BAC-guided assembly of cassava genomes has significantly improved the contiguity and accuracy of the repeat-rich *CMD2* region leading to the identification of additional sequences potentially relevant for virus resistance. A better resolution of the *CMD2* haplotypes will pave the way to a better understanding of resistance to cassava mosaic disease.

## Methods

### BAC sequencing

#### *Genomic libraries and bacterial artificial chromosome (BAC) sequencing*

BAC library construction and screening and BAC clone sequencing and assembly were performed by the INRA-CNRGV. High molecular weight (HMW) genomic DNA was prepared from young frozen leaves sampled on in vitro TME3 and cv. 60444 plantlets as described by Peterson et al. [48] and Gonthier et al. [49]. Agarose embedded HMW DNA was partially digested with HindIII (New England Biolabs, Ipswich, MA, USA), subjected to two-size selection steps by pulsed-field electrophoresis using a CHEF Mapper system (Bio-Rad Laboratories, Hercules, CA, USA). DNA was eluted, ligated into the pIndigoBAC-5 HindIII-Cloning Ready vector, and transformed into *Escherichia coli* electrocompetent cells. Pulsed-field migration programs, electrophoresis buffer, and ligation desalting conditions were performed based on Chalhoub et al. [50]. The insert size of the BAC clones was assessed using the FastNot I restriction enzyme (New England Biolabs, Ipswich, MA, USA) and analyzed by pulsed-field gel electrophoresis. Colony picking was carried out using a robotic workstation QPix2 XT (Molecular Devices, San José, CA, USA) using a white/blue selection. For each genotype, white colonies were arranged in 384-well microtiter plates containing LB medium with chloramphenicol (12.5 µg/mL) supplemented with 6% (v/v) glycerol (144 plates represented by 55,296 BAC clones for the TME3 genotype and 108 plates represented by 41,472 BAC clones for the cv. 60444 genotype). The resulting libraries represent ∼8.3-fold coverage of the TME3 genotype and ∼7.6-fold coverage of the cv. 60444 genotype. BAC clones were spotted on nylon membrane, screened with 27 radioactively ([α−33P]dCTP) labeled probes designed on the region of interest (Additional file 1: Table S1), and then analyzed with high-density filter reader program. The positive clones were verified by real-time PCR using the specific primers used for probe design. DNA were extracted from individual clones using Nucleobond Xtra midi kit (Macherey–Nagel, Düren, Nordrhein-Westfalen) and 2 µg of each individual BAC clone was used for PacBio library preparation.

Multiplexed SMRT® libraries using the standard Pacific Biosciences preparation protocol for 10 kb library with PacBio® Barcoded Adapters were prepared. Each library was then sequenced in one SMRT cell using the P6 polymerase with C4 chemistry. Sequencing was performed on a PacBio RS II sequencer. After a demultiplexing step, the sequence assembly was performed following the HGAP PacBio workflow [51] and using

the SMRT® Analysis (v2.3) software suite for HGAP implementation (https://github.com/PacificBiosciences/Bioinformatics-Training/wiki/HGAP). BAC end sequences confirmed the position of selected clones on the cassava genome. After the first round of PacBio sequencing of selected BACs, new probes were designed at both the ends of sequenced BACs. These probes were used to screen the Mes-TM3 and Mes-60444 BAC libraries again as described above. BAC-end Sanger sequencing was performed on the selected BACs to estimate the overlapping genomic regions between the sequenced and newly selected BACs. After several rounds of probe designing—BAC library screening—BAC-end sequencing—selection of overlapping BACs—BAC PacBio sequencing, BACs with appropriate overlapping regions were sequenced on the PacBio platform.

### Nanopore sequencing

Nanopore sequencing of cassava cv. 60444 and TME3 genomic DNA was performed using MinION ultra-long reads as described [30]. HMW DNA was extracted from young leaf tissues of cv. 60444 and TME3 cassava, grown at a 28 °C/25 °C day/night rhythm with a 16 h/8 h photoperiod, following the protocol provided by Oxford Nanopore Technologies, "High molecular weight gDNA extraction from plant leaves" downloaded from the ONT Community in October 2018, using QIAGEN Genomic tip 500/G (QIAGEN Cat. No. 10262). DNA quality was measured on 1% agarose gel and with a NanoDrop spectrophotometer. DNA was quantified using a Quantus Fluorometer (Promega). All the following steps involving handling DNA were performed with wide-bore pipet tips. To obtain ultra-long reads, the standard Rapid Adapters (RAD004) protocol (SQK-RAD004 Rapid Sequencing Kit, Oxford Nanopore Technologies (ONT), Oxford, UK) for genomic DNA was modified as described by Jain et al. [30]. MinION sequencing was performed as per manufacturer's guidelines using R9 flow cells (FLO-MIN106D, ONT) using a MinION sequencer Mk1B. Twelve flow cells were used for cv. 60444 ULR sequencing and seven for TME3; these flow cells generated 39,525 Mb (6,221,230 reads) and 41,340 Mb (6,549,320 reads) sequencing data for cv. 60444 and TME3, respectively, with 22,782 reads longer than 8 kb.

### Bionano sequencing

#### *Ultra high molecular weight DNA extraction*

To generate the optical map, ultra high molecular weight (UHMW) DNA were purified from 1 g of fresh dark treated very young leaves according to the Bionano Prep Plant Tissue DNA Isolation Base Protocol (30,068—Bionano Genomics) with the following specifications and modifications. Briefly, the leaves were fixed in fixing buffer containing formaldehyde. After 3 washes, leaves were cut in 2 mm pieces and disrupt with rotor stator in homogenization buffer containing spermine, spermidine, and beta-mercaptoethanol. Nuclei were washed, purified using a density gradient, and then embedded in agarose plugs. After overnight proteinase K digestion in lysis buffer (BNG) and 1-h treatment with RNAse A (Qiagen, MD, USA), plugs were washed 4 times in 1 × wash buffer (BNG) and 5 times in 1 × TE buffer (ThermoFisher Scientific, Waltham, MA). Then, plugs were melted for 2 min at 70 °C and solubilized with 2 µL of 0.5 U/µL AGARase enzyme (ThermoFisher Scientific, Waltham, MA) for 45 min at 43 °C. A dialysis step was performed in 1 × TE buffer (ThermoFisher Scientific, Waltham, MA) for 45 min to

purify DNA from any residues. The DNA samples were quantified by using the Qubit dsDNA BR Assay (Invitrogen, Carlsbad, CA, USA). Quality of megabase-size DNA was validated by pulsed-field gel electrophoresis.

### Data collection and optical map construction

Labeling and staining of the UHMW DNA were performed according to the Direct Label and Stain (DLS) protocol (BNG). Briefly, labeling was performed by incubating 750 ng genomic DNA with $1 \times$ DLE-1 enzyme (BNG) for 2 h in the presence of $1 \times$ DL-Green (BNG) and $1 \times$ DLE-1 buffer (BNG). Following proteinase K digestion and DL-Green clean-up, the DNA backbone was stained by mixing the labeled DNA with DNA Stain solution (BNG) in the presence of $1 \times$ flow buffer (BNG) and $1 \times$ DTT (BNG), and incubating overnight at room temperature. The DLS DNA concentration was measured with the Qubit dsDNA HS Assay (Invitrogen, Carlsbad, CA, USA).

Labeled and stained DNA was loaded on Saphyr chips. Loading of the chips and running of the BNG Saphyr System were all performed according to the Saphyr System User Guide. Digitalized labeled DNA molecules were assembled to optical maps using the BNG Access software.

### Genome assembly and BAC mapping

The assembly process was optimized with the reads from cv. 60444, using three different assemblers. *NanoFilt* [52] was used to generate a dataset with long reads (> 8 kb). *CANU* V2.3 [38], with the options stopOnLowCoverage = 5 and cnsErrorRate = 0.25, was used with all reads and on the > 8 kb read dataset. *Flye* V2.19.b1774 [39, 40], with default settings, was used with the same two datasets. *wtdbg2* V2.1 [37] was used on all reads with default settings and read length option set to 300, 2000, 3000, 5000, 8000, and 10,000. All final assemblies were polished using Illumina short reads, downloaded from the NCBI (SRX1393211 and SRX526747), with *pilon* V1.24 [53], using default settings, after mapping of the reads with *bwa mem* V0.7.17 [54] and *samtools* V1.13 [55].

The BAC sequences were mapped on the genomes using *Blasr* V5.1 [56]. *Blasr*, whose initial purpose is the mapping of corrected PacBio reads [56], was used on the full BAC sequences in fasta format, with default settings. Only the longest alignment per BAC was used to compute the percentage of BAC mapping on the genome. The same approach was used to compare public cassava genomes with our best assemblies. The results of BAC mapping are available in Additional file 1: Table S2. The quality of the assemblies was assessed with *BUSCO* V5.3.0 [42], using auto-lineage settings. The BAC mapping showed that the *CMD2* region was better resolved using Flye with default settings and reads longer than 8 kb. Based on BUSCO scores, the best assembly of cv. 60444 was generated with CANU [38] using all reads (Additional file 1: Table S2). However, this assembly, better in completeness, was outperformed by the one generated with Flye using reads above 8 kb when assessed by BAC mapping.

The TME3 reads were filtered with *NanoFilt* [52] to generate the > 8 kb read dataset. The assembly of TME3 reads was done using the best approaches as previously assessed by BAC mapping on cv. 60444 assemblies (i.e., *Flye* with default settings using all reads and reads longer than 8 kb). The selected TME3 assembly was polished using Pilon V1.24 [53]. The BAC mapping was performed by *Blasr* [56].

Bionano optical mapping was used to scaffold the selected cv. 60444 and TME3 genomes (i.e., *Flye* with default setting using > 8 kb reads). Genomic statistics on these final genomes, and on public genomes, were computed using QUAST [57], with default settings. *BUSCO* V5.3.0 [42], with auto-lineage settings, was used to estimate completeness and duplication (based on duplication value reported by BUSCO of its single gene marker set).

### Marker mapping

The 64 genetic markers from Rabbi et al. [15] were mapped on the *CMD2* haplotypes produced in this study and previously available in public databases [15] using blastn V2.10.0 [58] with an *e* value cut-off of 10e − 3. The top hit was taken to determine the position of each marker on the chromosome.

The *CMD2* locus were mapped using blastn (BLAST 2.9.0 +) searches of 101 markers on several genomes of cassava (60444 H1 and H2 GCA_963409065.1 Cornet, TME3 H1 and H2 GCA_963409055.1 Cornet, TME3 GCA_003957995.1 Kuon, and 60444 GCA_003957885.1 Kuon). These 101 markers (Additional file 1: Table S6) included 6 classical markers (RFLP and SSR markers) published by Akano et al. [32], Lokko et al. [33], Okogbenin et al. [34], and Okogbenin et al. [36], 31 SNP markers on chromosome 12 from Rabbi et al. [17] (the 31 markers were selected by taking the SNP with the highest − log *p* value plus 15 markers upstream stream and 15 markers downstream of this peak SNP marker), and 64 other SNP markers published by Wolfe et al. [16].

### Genome annotation

#### *Repetitive elements and noncoding RNA annotation*

We used two strategies to predict the repeat sequences in the two genotypes. The first strategy was ab initio prediction. RepeatModeler v2.0 (http://www.repeatmasker.org/RepeatModeler/) and LTR_FINDER v1.0.7 [59] were employed to generate custom libraries. Then, the assembled sequences of these two genotypes were mapped against the libraries, respectively, to generate the ab initio annotation results. The second strategy was a homology-based prediction. The assembled sequences were aligned to the RepBase v23.05 ((http://www.girinst.org/repbase/) by using RepeatMasker v4.1.0 [60]. Then, the information from the above two methods was integrated into non-redundant results.

Transfer RNAs (tRNAs) were predicted by tRNAscan-SE v1.43 [61] with default parameters. To predict ribosomal RNAs (rRNAs), the genome assemblies were aligned against the RNA families (Rfam) v14.1 database [62] by the Blastn program [58].

#### *Gene annotation*

Three different strategies were used to predict the gene set of the two cassava genotypes:

> *Homology annotation.* We aligned the protein sequences from five published genomes, including *Manihot esculenta* (AM560-2 V8) [22], *Ricinus communis* (https://bioinformatics.psb.ugent.be/plaza/versions/plaza_v4_5_dicots/), and *Jatropha curcas* (NCBI accession: GCA_000696525.1), respectively, against our assem-

blies to predict genes based on homology. The potential homology-based genes were searched by GeMoMa v1.7 [63].

*RNA-seq annotation.* We removed the redundancies in a public Iso-seq cassava data-set [41] using Cupcake v12.1.0 (https://github.com/Magdoll/cDNA_Cupcake/) and got the unique isoforms that were used as input for the Assemble Spliced Alignments (PASA) pipeline v2.4.1 [64]. For Illumina RNA-seq data, SRR25338832, Trinity [65] was used to assemble the data. Then, we used PASA to identify the potential gene structures.

De novo *annotation.* In order to discard pseudo gene predictions, we used "N" to substitute the repetitive elements of the assembled sequences. Then, we extracted the complete genes with multiple exons and start/stop codons from the predicted genes of homologous annotation and RNA-seq annotation to build the training set for the do novo gene predictors. We identified 7553 complete genes in cv. 60444 and 7394 complete genes in TME3 to construct the training set, respectively. Subsequently, AUGUSTUS v3.3.3 [66], SNAP v2006-07–28 [67], and GlimmerHMM v3.0.4 [68] were trained based on the training set and then operated to identify the potential gene models. GeneMark-ES v4.57_lic [69] was executed to predict genes with default parameter.

For the integration of the annotation results, we employed the Evidencemodeler v1.1.1 [70] to generate non-redundant and comprehensive gene sets. After that, we used the PASA pipeline again to correct the potential errors to generate the final gene sets for the two genotypes. Then, we used BUSCO v5.0.0 [42] and the embryophyta_odb10 dataset to assess the quality of the predicted gene sets.

Functional annotation of the predicted genes was operated by running BlastP [71] setting an *e* value cut-off of 1e-05 against the public protein function databases Uniprot/SwissProt [72] and NCBI NR [73] (RefSeq non-redundant protein record). Gene Ontology (GO) and Kyoto Encyclopedia of Genes and Genomes (KEGG) terms were classified using eggNOG v2.1.6 [74].

### Resistance gene

The *CMD2* SNP markers of Rabbi et al. [15] and Rabbi et al. [17] were used to define the *CMD2* region on our haplotypes. All proteins of this region were then analyzed with prPred [75] to predict the probability that these proteins contributed to biotic stress resistance. Eighty-one proteins with a probability above 0.1 were retained for both haplotypes of TME3.

The 81 proteins were searched in the cv. 60444 genome (GCA_963409065.1 Cornet), the NCBI TME3 genome (GCA_003957885.1 Kuon), and IsoSeq data (personal communication from Kuon et al. [41]) of the two genotypes by orthologous enrichment with the program Forty-two V0.1416 [76, 77]. The generated orthologous group (OG) files were aligned with mafft [78], with default settings. The alignments were then cleaned of mis-predicted stretches with HMMCleaner [79], using leave-one-out profiles, with c1 =0.40 and default weights for c2 to c4. Individual phylogenetic trees were produced with Phyml v3.1, through seaview [80], and subtrees with orthologous sequences were manually created. The presence of the 81 proteins in genomes and Isoseq data were determined based on presence in

these subtrees. The 81 proteins were also analyzed by Interproscan V5.48–83 [81]. CMD2 regions and corresponding proteins are available in the figshare repository [82].

## Supplementary Information

---

Additional file 1: Figs. S1–S2 and Tables S1–S6.

Additional file 2: Peer review history.

---

### Acknowledgements

We thank David Colignon and the CÉCI for help with computing cluster usage. The authors thank Ismail Rabbi (IITA) for data sharing and fruitful discussions.

### Review history

The review history is available as Additional file 2.

### Peer review information

### Authors' contributions

LC, SZ, JL, YN, MH, YVdP, and HV conceived the study and wrote the initial manuscript. SS, SZ, and LC produced ONT data. SS and SZ produced BACs. LC and LM ran bioinformatics analysis. LC and YN made figures. CC and WM produced optical maps and corrected genomes. JL, SR, and YVdP made protein predictions. LC ran resistance protein analysis. All authors read and accepted the final manuscript.

### Funding

The authors acknowledge financial support from the Belgian FNRS grant M.i.S. F.4515.17 to H.V. and grant 1.B456.20 to S.S.Z. and H.V.YVdP acknowledges funding from the European Research Council (ERC) under the European Union's 501 Horizon 2020 research and innovation program (No. 833522) and from Ghent University (Methusalem 502 funding, BOF.MET.2021.0005.01).The authors acknowledge financial support from the Belgian FNRS grant M.i.S. F.4515.17 and J018721F  to H.V. and grant 1.B456.20 to S.S.Z. and H.V. HV acknowledges internal C1 funding from KU Leuven (3E21053). YN acknowledges funding from the Fonds Wetenschappelijk Onderzoek (FWO) Vlaanderen grant no. 1SHEQ24N.

### Data availability

L Cornet and H Vanderschuren have summitted raw data to the National Center for Biotechnology Information (NCBI) under the Bioproject accession no. PRJNA981703 [83], Raw nanopore reads no. SRR25338822 for cv. 60444 and no. SRR25338821 for TME3, bionano optical maps accession no. SUPPF_0000005665 for cv. 60444 and no. SUPPF_0000005666 for TME3 (both available from https://ftp.ncbi.nlm.nih.gov/pub/supplementary_data/bionanomaps.csv), and RNAseq reads accession no. SRR25338832. L Cornet and H Vanderschuren have summitted genome assemblies to European Nucleotide Archive (ENA) under the Bioproject PRJEB65447, genome accession GCA_963409065.1 for cv. 60444 and GCA_963409055.1 for TME3.

## Declarations

### Ethics approval and consent to participate

Not applicable.

### Competing interests

The authors declare no competing interests.

### Author details

[1]Plant Genetics and Rhizosphere Processes Laboratory, TERRA Teaching and Research Center, Gembloux Agro-Bio Tech, University of Liège, Gembloux Agro-Bio TechGembloux, Belgium. [2]Department of Plant Biotechnology and Bioinformatics, Ghent University and VIB Center for Plant Systems Biology, Ghent, Belgium. [3]Laboratory of Tropical Crop Improvement, Division of Crop Biotechnics, Biosystems Department, KU Leuven, Leuven, Belgium. [4]InBioS, PhytoSYSTEMS, Eukaryotic Phylogenomics, University of Liège, PhytoSYSTEMS, Liège, Belgium. [5]CNRGV, Centre National de Ressources Génomiques Végétales, Toulouse, France. [6]InBioS, PhytoSYSTEMS, Translational Plant Biology, University of Liège, Liège, Belgium. [7]Centre for Microbial Ecology and Genomics, Department of Biochemistry, Genetics and Microbiology, University of Pretoria, Pretoria 0028, South Africa. [8]College of Horticulture, Academy for Advanced Interdisciplinary Studies, Nanjing Agricultural University, Nanjing, China. [9]KU Leuven Plant Institute (LPI), Kasteelpark Arenberg 31, Leuven, Heverlee, 3001, Belgium.

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

## 