## [Additional file 2: Peer review history. · Genome Biology]

Review history

First round of review

Reviewer 1

The short report of Luc et al. describes the construction of a locally haplotype-resolved sequence assembly of two cassava genomes with the goal of assisting the genetic mapping of resistance to a viral disease. To obtain the assembly, they used whole-genome Oxford Nanopore (ONT) sequence data and PacBio HiFi data of bacterial artificial chromosomes (BACs). The manuscript is concise, well-written and easy to follow. The display items are of high quality. I am uncertain whether this manuscript is of sufficiently broad interest.

Specific comments:

1. From Genome Biology's instructions for authors: "Genome Biology publishes Short Reports that are concise studies of high quality and broad interest. Short Reports can present new research findings, or can present a new method or software." I believe the definition of more putative candidate genes thanks to the improved assembly of the CMD2 locus is highly valuable to the small set of researchers that work on cloning this gene, but broader interest would be elicited only by the cloning and functional validation of the CMD2 gene or genes.
2. So the broad interest of the present manuscript hinges on the relevance of the BAC-guided assembly approach to a wider audience. The authors "screened BAC libraries using high-density filter hybridization with radioactively labeled probes." This is a tried and true method to identify BACs carrying markers of interest. But I'm worried that the construction and handling of BAC libraries is a laborious approach that many researchers would wish to avoid unless no other alternatives were available. Owing to large laborious nature of BAC mapping, it also does not scale to pangenomics, the analysis of contiguous genome sequences for many individuals of a species. So the BAC-guided approach would be limited to well-resourced gene isolation project that deal with a small and predefined set of genotypes. Still, even within this circumscribed remit they may be broad interest because gene isolation is an endeavour pursued by many scientists.
3. Now the question remains if there are no easier alternatives to the approach propounded by authors, especially when taking into account the latest progress in DNA sequencing technology. The authors consider ONT reads that are longer than 8 (in word: eight) kb as "ultra-long", whereas many other researcher set that threshold at 50 (fifty) kb. This is an important point as you can expect to resolve recent insertion of common classes of transposable elements in plant genomes with 50 kb reads, but not with 8 kb. I hope I'm not being unfair to the authors but a plausible explanation for this improbable choice of threshold that occurred to me is that the authors were working with ONT data that were generated several years ago. They stated that the method for DNA extraction followed a protocol released in 2018 (p. 9, l. 54). ONT sequencing has improved a lot since then years, and using newer data with longer reads may results in a better whole-genome assembly. Ideally, a methods paper should report on, or even next years' methods. I believe the authors have to show a whole-genome sequence assembly with a sufficient number of ONT reads longer than 50 kb is unable to resolve the CMD2 loci in a haplotype-resolved manner in order to show that BAC-guided local assembly is truly needed.
4. But the most important blind spot of the present manuscript is the omission of whole-genome accurate long-read sequencing data generated on the PacBio HiFi sequencing platform. Many impressive reports on the assembly of chromosome-scale, haplotype resolved

genome sequences of highly heterozygous and even autopolyploid crops have been published in recent years, e.g. on apple (doi: 10.46471/gigabyte.69), potato (doi: 10.1186/s13059-023-03160-z) and sugarcane (doi: 10.1038/s41586-024-07231-4). These studies employed PacBio HiFi sequencing together with chromosome-conformation capture sequencing (Hi-C) or genetic mapping. HiFi and Hi-C are scalable and offered by commercial vendor, very much in contrast to BAC handling. Since cassava has a small genome (~1.5 Gb diploid genome size), I believe it is not unreasonable to ask the authors to generate whole-genome PacBio HiFi data. If they were then able to show that the assembly of these data with state-of-the-art assembly algorithms cannot resolve the CMD2 locus, they would have made a convincing case for why BAC-guided local assembly is needed and provided an important case study into the limitations of current approaches for whole-genome sequence assembly.

Reviewer 2

I am not qualified to assess the statistical/computational analyses involved in this work. However it appears to be technically excellent and easy to understand. I do however feel that the authors need to do a better job of explaining the importance of cassava and its pests respectively since this will not be readily apparent to many readers of the article. Beyond this I found it to be an excellent manuscript which is of the standard I would expect from this journal.”

Authors' response to reviewers

Reviewer 1

1. From Genome Biology's instructions for authors: "Genome Biology publishes Short Reports that are concise studies of high quality and broad interest. Short Reports can present new research findings, or can present a new method or software." I believe the definition of more putative candidate genes thanks to the improved assembly of the CMD2 locus is highly valuable to the small set of researchers that work on cloning this gene, but broader interest would be elicited only by the cloning and functional validation of the CMD2 gene or genes.

The manuscript aims at presenting an improved assembly of a genomic region (associated with geminivirus resistance), the interest of which goes beyond the cassava community. Indeed, geminiviruses are infectious to many plant species including several major crops (i.e., tomato, wheat, cotton, maize, bean, cucurbits, sugarbeet, etc.). Therefore, improved assemblies of a locus that has been associated with geminivirus resistance would be of high interest to the plant virology community at large. Because the biology and host response to circular DNA viruses, as well as the natural tolerance, remain poorly characterized, advancing the identification of potential candidates for resistance in a natural geminivirus pathosystem is of high relevance for the plant biology/virology community. The progress in assembling the complex and highly repetitive CMD2 region (the best assembled to date), as well as the identification and reporting of 81 resistance-related proteins present in the CMD2, is paving the way for the characterization of geminivirus resistance which is key for the community. In the revised manuscript we have also highlighted the importance of cassava and its associated natural geminivirus pathosystem, further justifying the importance of the study. Moreover, our study presents an original optimisation of assemblies using BAC sequencing. This has not been reported before with HTS data for plant genomes. We trust this provides an interesting contribution to the community by establishing a valuable approach for optimization and validation of genome assemblies.

Furthermore, Genome Biology has recently published several Short Reports leading to

substantial improvements on very specific topics: for example, Wang et al., 2022 studied the link between thermotolerance and RNA methylation in Arabidopsis (<https://genomebiology.biomedcentral.com/articles/10.1186/s13059-022-02814-8>), while McIntyre et al., 2024 identified 54 orthologous proteins of the XIST interactome, linked to X chromosome inactivation in the opossum (<https://genomebiology.biomedcentral.com/articles/10.1186/s13059-024-03280-0>). Therefore, in comparison with these papers, we believe our paper also makes a perfect fit for Genome Biology.

2. So the broad interest of the present manuscript hinges on the relevance of the BAC-guided assembly approach to a wider audience. The authors "screened BAC libraries using high-density filter hybridization with radioactively labeled probes." This is a tried and true method to identify BACs carrying markers of interest. But I'm worried that the construction and handling of BAC libraries is a laborious approach that many researchers would wish to avoid unless no other alternatives were available. Owing to the laborious nature of BAC mapping, it also does not scale to pangenomics, the analysis of contiguous genome sequences for many individuals of a species. So the BAC-guided approach would be limited to well-resourced gene isolation projects that deal with a small and predefined set of genotypes. Still, even within this circumscribed remit they may be of broad interest because gene isolation is an endeavour pursued by many scientists.

We understand the concern of the reviewer, but would like to stress again that the objective of our BAC-guided approach was to enhance specific regions of the genome—those that are highly complex and of significant interest—with BAC sequences of confirmed reliability, assembled with high coverage using PacBio RS. This strategy results in the selection of the best 'local optima' for assembly rather than to strive for 'the best overall' genomes, an approach typically used in pangenomics. Our optimization procedure was instrumental to build confidence in the assembled contigs outside the BAC sequences which should be considered as the ultimate quality of genome sequence contiguity.

In summary, our method achieves the best assembly to date for the CMD2 region, which remains an important target and result because of its complexity as stated above. Although we could extrapolate that several potentially complex regions of the cassava genome have been improved based on the results of the CMD2 assembly, a similar methodology to select the best genome would indeed involve a random sampling of genomic sites to validate the superiority of the full genome assembly. However, such an approach would indeed be too labour-intensive and the objective here was to focus on specific regions of interest, which makes the approach feasible. We also acknowledged the point mentioned by the reviewer in the Methods section in which we report BUSCO scores to compare the full genome assemblies.

3. Now the question remains if there are no easier alternatives to the approach propounded by authors, especially when taking into account the latest progress in DNA sequencing technology. The authors consider ONT reads that are longer than 8 (in word: eight) kb as "ultra-long", whereas many other researchers set that threshold at 50 (fifty) kb. This is an important point as you can expect to resolve recent insertion of common classes of transposable elements in plant genomes with 50 kb reads, but not with 8 kb. I hope I'm not being unfair to the authors but a plausible explanation for this improbable choice of threshold that occurred to me is that the authors were working with ONT data that were generated several years ago. They stated that the method for DNA extraction followed a protocol released in 2018 (p. 9, l. 54). ONT

sequencing has improved a lot since then years, and using newer data with longer reads may results in a better whole-genome assembly. Ideally, a methods paper should report on, or even next years' methods. I believe the authors have to show a whole-genome sequence assembly with a sufficient number of ONT reads longer than 50 kb is unable to resolve the CMD2 loci in a haplotype-resolved manner in order to show that BAC-guided local assembly is truly needed.

We appreciate the comment of the reviewer, but as mentioned in the previous comment, the BAC mapping technique proposed here is not designed to select the best genome overall but rather to identify the best local optimum for a specific region of high importance, here, virus resistance. While our data indeed date back to a few years ago, and flow cell technology has improved since then, the significant improvement of the CMD2 region, as demonstrated by the mapping of marker genes using shorter reads than those available today, underscores the effectiveness of our approach, even with shorter read sequences.

As our focus is on validating a specific region of assembly, and not competing with the latest sequencing technologies, we feel that producing a new sequencing dataset with longer reads would not disprove our approach, although it might indeed also lead to better assemblies. Indeed, the resources required for such a re-sequencing effort would not bring substantial new insights into the efficiency of this targeted approach. However, this does not detract from the method itself, as its primary purpose is to improve and validate genomic regions based on highly reliable BAC sequences.

Importantly, because BAC sequences are also available in different plant research communities, the proposed methodology could also be applied to the so-called orphan crops which structurally lack the means to always use the latest technologies to generate assemblies of genomic regions with increased accuracy and contiguity. Therefore, our method would also find immediate application for genomes of orphan crops for which sequencing data not generated with the latest technologies could be used to generate improved assemblies of specific regions.

4. But the most important blind spot of the present manuscript is the omission of whole-genome accurate long-read sequencing data generated on the PacBio HiFi sequencing platform. Many impressive reports on the assembly of chromosome-scale, haplotype resolved genome sequences of highly heterozygous and even autopolyploid crops have been published in recent years, e.g. on apple (doi: 10.46471/gigabyte.69), potato (doi: 10.1186/s13059-023-03160-z) and sugarcane (doi: 10.1038/s41586-024-07231-4). These studies employed PacBio HiFi sequencing together with chromosome-conformation capture sequencing (Hi-C) or genetic mapping. HiFi and Hi-C are scalable and offered by commercial vendor, very much in contrast to BAC handling. Since cassava has a small genome (~1.5 Gb diploid genome size), I believe it is not unreasonable to ask the authors to generate whole-genome PacBio HiFi data. If they were then able to show that the assembly of these data with state-of-the-art assembly algorithms cannot resolve the CMD2 locus, they would have made a convincing case for why BAC-guided local assembly is needed and provided an important case study into the limitations of current approaches for whole-genome sequence assembly.

We agree with the reviewer that the PacBio HiFi sequencing technology, with its high accuracy, is indeed an interesting approach. Nevertheless, this technology remains very expensive, in particular for crop species such as cassava. As stated in the manuscript, the objective was to improve the assembly of CMD2 region by using a BAC-based validation approach and

affordable technologies at the time of study. We clearly demonstrate in the manuscript that our CMD2 assembly outcompetes the available ones including those generated with high coverage of PacBio long reads and Hi-C data. Furthermore, it is important to note that the PacBio HiFi technology still does not allow for the perfect assembly of a genome. For instance, Rabanal et al. (2022) (<https://pmc.ncbi.nlm.nih.gov/articles/PMC9757041/>) clearly demonstrated that managing GA/TC repeats remains problematic even with HiFi reads.

In conclusion, we believe that our methodology provides a real added value to the assembly of repetitive regions, even when using HiFi data, making our BAC-based approach a unique and useful method.

Reviewer 2

I am not qualified to assess the statistical/computational analyses involved in this work. However it appears to be technically excellent and easy to understand.

1. I do however feel that the authors need to do a better job of explaining the importance of cassava and its pests respectively since this will not be readily apparent to many readers of the article. Beyond this I found it to be an excellent manuscript which is of the standard I would expect from this journal

We thank the reviewer for his/her constrictive comments and as requested by the reviewer, we have now added a paragraph on the importance of cassava and cassava mosaic disease to the first section of the manuscript (highlighted in blue color).

Second round of review

Reviewer 1

The authors have written a considerate response to my concerns. It is now easier to understand why they conducted the study in the way they did. I still have some doubts about the wider impact of this study given the rapid technical advances in DNA sequencing. But as referee 2 has pointed out this is a well-written manuscript on a sound piece of research, so there are no technical concerns that would militate against publication.